# The effects on clinical trial activity of direct funding and taxation policy interventions made by government: A systematic review

**Sam Crosby**[1]⊕*, **Esther Rajadurai**[2]‡, **Stephen Jan**[1]‡, **Bruce Neal**[1]⊕, **Richard Holden**[3]

**1** The George Institute for Global Health, Newtown, NSW, Australia, **2** The McKell Institute, Sydney, NSW, Australia, **3** UNSW Business School, UNSW, Kensington, NSW, Australia

⊕ These authors contributed equally to this work.
‡ These authors also contributed equally to this work
* scrosby@georgeinstitute.org.au

**Data Availability Statement:** All relevant data are within the paper and its Supporting Information files.

## Abstract

### Context

Governments have attempted to increase clinical trial activity in their jurisdictions using a range of methods including targeted direct funding and industry tax rebates. The effectiveness of the different approaches employed is unclear.

### Objective

To systematically review the effects of direct government financing interventions by allowing companies to reduce their tax payable on clinical trial activity.

### Data sources

Pub Med, Scopus, Sage, ProQuest, Google Scholar and Google were searched up to the 11th of April 2022. In addition, the reference lists of all potentially eligible documents were hand searched to identify additional reports. Following feedback from co-authors, information on a small number of additional interventions were specifically sought out and included.

### Data extraction

Summary information about potentially eligible reports were reviewed independently by two researchers, followed by extraction of data into a structured spreadsheet for eligible studies. The primary outcomes of interest were the number of clinical trials and the expenditure on clinical trials but data about other evaluations were also collected.

### Results

There were 1694 potentially eligible reports that were reviewed. Full text assessments were done for 304, and 30 reports that provided data on 43 interventions were included– 29 that deployed targeted direct funding and 14 that provided tax rebates or exemptions. There were data describing effects on a primary outcome for 25/41 of the interventions. The most common types of interventions were direct funding to researchers via special granting

**Funding:** The author(s) received no specific funding for this work.

**Competing interests:** The authors have declared that no competing interests exist.

mechanisms and tax offsets to companies and research organisations. All 25 of the studies for which data were available reported a positive impact on numbers and/or expenditure on clinical trials though the robustness of evaluations was limited for many. Estimates of the magnitude of effects of interventions were reported inconsistently, varied substantially, and could not be synthesised quantitatively, though targeted direct funding interventions appeared to be associated with more immediate impact on clinical trial activity.

## Conclusion

There is a high likelihood that governments can increase clinical trial activity with either direct or indirect fiscal mechanisms. Direct funding may provide a more immediate and tangible return on investment than tax rebates.

## Introduction

Randomized clinical trials are gold standard research investigations designed to generate high quality data about ways to prevent, detect or treat medical conditions [1]. If done well, the evidence that is derived from clinical trials forms the basis for the implementation of new health interventions, clinical guidelines, and government policy. Globally the clinical trial industry was valued at between $44.3 billion [2] and $51.25 billion [3] (USD) in 2020; the USA accounting for nearly half this at $21 billion (USD) while in Australia the sector is worth around $1 billion (AUD). Clinical trials have become an important source of employment and external investment for many jurisdictions [4], as well as providing a means for the community to access novel therapies earlier. The rapid development and testing of multiple COVID_19 vaccines throughout 2020 has reinforced the importance of maintaining institutional capacity to support clinical trials.

Governments around the world have pursued a range of interventions to generate more local clinical trial activity and to attract clinical trials from other jurisdictions. Low- and middle-income country governments, for example, have built clinical trial infrastructure to encourage overseas investment that leverages access to their large populations and low-cost base. In India, for example, the government established training institutes to upskill the clinical trial workforce and hospital departments were provided additional equipment and funding for staff to act as clinical trial investigators [5,6]. These types of infrastructure investments have increased activity though the resulting trials may prioritise the health concerns of the higher-income markets funding the trials over local health issues. At the same time, in developed countries, over-complex infrastructure and fragmented regulatory and approval systems have been a frequent target of reform. In addition, saturation of the clinical trials services market has meant that fees for some aspects of trial conduct have risen to unsustainable levels, leading to reduced investment and activity particularly in non-commercial trials. A 2013 Government of Australia review found that "Australia has become one of the most expensive locations for clinical trials in the world" [7] spurring calls to make the sector more efficient and more internationally competitive.

There are a variety of different mechanisms available to governments seeking to intervene in the sector and promote growth in clinical trial activity. The range of options includes streamlining processes, investing in infrastructure, waiving fees and charges, directly funding researchers, making advanced market commitments and manipulating tax instruments. The objective of this paper was to systematically review evidence of interventions that have sought to increase clinical trial activity by special direct funding schemes or tax policy mechanisms.

Ongoing general funding for higher education, medical research and the healthcare system was outside the scope of this review. As such, direct funding interventions were included in the review only if they provided significant new resources targeted specifically to clinical trials.

## Methods

This systematic review was conducted in accordance with the Cochrane Handbook for Systematic Reviews of Interventions [8]. The guiding question was: 'What are the effects of government's actions targeting funding and taxation regimes on clinical trial activity and investment?' The protocol was registered with the International Prospective Register of Systematic Reviews (PROSPERO) under registration number CRD42020191510 as a slightly broader question of 'What are the effects of governments actions on clinical trial activity?'. Other government actions such as the improvement of ethics and governance systems will be addressed separately in another paper.

### Search strategy

The search strategy was developed in consultation with the UNSW Library research service where key search terms were identified ("clinical trials" and "public policy" as free text key words). These terms were combined using the Boolean operator 'AND' to complete searches of Pub Med, Scopus, Sage, ProQuest and Google Scholar databases on all reports up to 11 April 2022. This was followed by a search of the internet for grey literature done using similar terms in the search engine Google. Finally, a hand search of the references of all included reports was done and co-authors were consulted. No time constraints or language barriers were placed on the search parameters.

The reports identified from the searches of Pub Med, Scopus, Sage and ProQuest were exported to Covidence, which automatically removed duplicate entries. The reports from Google Scholar were exported to Publish or Perish. The Google search engine results as well as the reports identified from the hand searches of reference lists were recorded in an Excel spreadsheet and duplicates were excluded by hand. Additionally, after feedback from co-authors, additional targeted research was done into a small number of interventions that were subsequently included.

### Study inclusion criteria

Studies were eligible for inclusion if they (1) reported on a policy intervention of interest (new direct funding or taxation targeting clinical trial activity); (2) provided some measure of the impact of the intervention on clinical trial activity; and (3) the intervention was implemented by a national or sub-national jurisdiction. Studies that analysed a jurisdiction's clinical trial sector or the funding or taxation systems but did not report on a specific intervention were excluded. Studies that identified the implementation of an eligible intervention but failed to report on an outcome of interest were recorded in the listings but noted to have missing outcome data. In addition, studies that reported on established prerequisites for clinical trials such as regular research granting schemes, existing health system support structures for clinical trials, or ongoing advanced clinical trial-focused education systems were not included unless a significant modification to the existing program was reported upon.

### Study selection

Two authors (SC and ER) independently screened all potentially eligible studies. For the studies identified from Pub Med, Scopus, Sage and ProQuest this comprised an initial review of titles and abstracts with review of the full text articles done only for those that passed initial

screening. For the studies identified from Google Scholar and using the Google search engine the screening was a single step process. Where one reviewer included or excluded a study in contradiction to the second reviewer a discussion was had, and consensus was reached about whether the study was eligible.

## Data extraction

Two authors (SC and ER) independently extracted data from each eligible study into separate copies of the same spreadsheet. Once both authors had completed the data extraction process every item of data was compared and discrepancies were reconciled by discussion. The study characteristics extracted were: country, year of publication, intervention (direct funding or tax rebate), impact of each intervention on main outcomes of interest (number of trials, expenditure on trials) and other reported outcomes.

## Quality assessment

As non-randomized intervention studies, the quality of each study was assessed by 4 parameters as advised by the Cochrane Handbook for Systematic Reviews [8]. The 4 parameters that the studies were assessed against were 'confounding bias' that arises when there are systematic differences between experimental intervention and comparator groups, which represent a deviation from the intended interventions; 'selection bias' that arises when later follow-up is missing for individuals initially included and followed, bias due to exclusion of individuals with missing information about intervention status or other variables such as confounders; 'information bias' introduced by either differential or non-differential errors in measurement of outcome data; and 'reporting bias' representing selective reporting of results from among multiple measurements of the outcome, analyses or subgroups in a way that depends on the findings.

## Categorisation of interventions

The interventions were categorised as "direct funding [9–28]" or "taxation policy [5,12,16,20,22,23,26,29–37]". Direct funding referred to governments or health departments directly expending funds on programs aimed at supporting or facilitating trial activity, while taxation policy referred to forgone revenue through tax credits or exemptions designed to encourage expenditure by a third party such as a pharmaceutical company or other research organisation. Those interventions categorised as "direct funding" were further divided into the sub-categories of: (1) funding for clinical trial infrastructure [5,9,12] (where research infrastructure such as laboratories or databases necessary for doing clinical trials were built); (2) funding for private companies [28,38] (where a company's research was directly funded); (3) funding for patient participants [13–16] (where the cost of the trial borne by the patient was covered by a government program); (4) funding for researchers (where significant grants were awarded directly to public and private researchers either outside or in addition to standard funding programs); [15–24,38–41] (5) funding for workforce development (where governments funded specific upskilling programs to develop a workforce necessary to conduct clinical trials); [5,6,25,27] or (6) advanced market commitment where the Government guaranteed purchase at a set price if the technology could be proved in the trial [10,28].

Interventions categorised as "taxation policy" were divided into three different categories; (1) research and development tax credits or offsets (where the research and development costs of companies were deducted from their revenue); [12,16,20,22,23,26,29–36] (2) fees and charges exemptions (where specific fees and charges related to clinical trials were removed); [5,29,37] or (3) preferential income tax rates (where companies engaged in clinical trials were taxed at a lower rate than other companies) [36].

## Outcomes

The primary outcomes of interest were the number of clinical trials. Secondary outcomes were financial impact and community access to quality healthcare. Community access to quality healthcare was discontinued as an outcome since there was little reporting on this outcome. 'Financial impact' was measured as expenditure on clinical trials which was defined as funding for trial activity from any source but the data of primary interest to governments was that related to expenditure on clinical trials by multinational healthcare companies.

## Data synthesis

To enable the effects of interventions on each outcome to be summarized, the effect of each intervention on each outcome was documented as positive (when a favourable impact was identified), null (when no impact was identified), adverse (when a negative effect was identified) or missing. In some instances where more comprehensive evaluation of the intervention was available (ROI, or cost-benefit) this was also included. The numbers of studies reporting each form of outcome was summarized.

# Results

## Identified studies

There were a total of 1694 potentially relevant reports identified in the database searches (S1 Fig). 205 reports were retrieved from peer reviewed databases and examined in Covidence. A further 9829 were identified from Google Scholar and ~1,400,000 from the Google search engine. These searches were restricted to the first ~2000 hits in each since the yield of potentially relevant studies fell rapidly. Google search outcomes were reviewed and recorded in Excel. Of these reports, 1145 and 223 were included through Google Scholar and Google respectively as potentially relevant. The bibliographies of potentially relevant texts were also reviewed resulting in an additional 121 potentially relevant reports identified from other sources. Of the 304 reports assessed in full text format there were 276 excluded as failing to meet the inclusion criteria. This left 28 reports with data describing 40 distinct interventions. Based on feedback from co-authors additional targeted searches were performed to gain information on three interventions described in two reports. This increased the number of reports to 30 and the number of interventions to 43. These interventions included 29 targeting direct funding and 14 targeting taxation policy. 15 of these reports were published since 2015, 7 between 2005–2015 and 8 before 2005 (Appendix 1).

All reports for which data were available included some form of 'before-after comparison', mostly with little formal description of methodology. The background settings within which the different interventions were tested varied considerably across the studies. The quality assessment of the included papers (Table 1) identified the majority as being at a high or moderate risk of bias. This finding was consequent upon both the underlying weakness for many research designs and the incomplete reporting of the information required to make a comprehensive assessment of the risk of bias.

## Characteristics of the interventions and the available outcome data (Tables 3 & 4)

The interventions were distributed across 12 countries and jurisdictions (S2 Fig). The country with the most interventions was the USA (8 direct funding interventions and 2 tax policies) followed by India (3 direct funding interventions and 3 tax policies).

**Table 1. Quality assessment of selected texts.**

| Not Applicable | | Low Risk | Medium Risk | High Risk |
|---|---|---|---|---|

| Paper (by lead author) | Year Published | Confounding Bias | Selection Bias | Information Bias | Reporting Bias |
|---|---|---|---|---|---|
| ACoSaQiH | 2020 | Low Risk | Medium Risk | Medium Risk | Medium Risk |
| AusTrade | 2018 | High Risk | High Risk | High Risk | High Risk |
| Blume-Kohout | 2009 | Low Risk | Low Risk | Low Risk | Low Risk |
| Blume-Kohout | 2012 | Low Risk | Low Risk | Low Risk | Low Risk |
| Cheng | 2007 | Medium Risk | High Risk | Medium Risk | High Risk |
| Chit | 2018 | Low Risk | Low Risk | Medium Risk | Medium Risk |
| Chinnery | 2021 | Medium Risk | High Risk | Low Risk | Medium Risk |
| Choudhury | 2019 | Medium Risk | Medium Risk | Low Risk | Medium Risk |
| Christakis | 1989 | High Risk | High Risk | Low Risk | High Risk |
| Davies | 2016 | High Risk | Low Risk | Medium Risk | High Risk |
| De Padua Risolia | 2015 | Low Risk | Medium Risk | Medium Risk | Low Risk |
| Hirst | 1991 | Medium Risk | Low Risk | Medium Risk | Medium Risk |
| Hudson | 2016 | Medium Risk | Medium Risk | Medium Risk | High Risk |
| Iizuka | 2016 | Low Risk | Medium Risk | Medium Risk | Low Risk |
| Kim | 2021 | Low Risk | Low Risk | Medium Risk | Medium Risk |
| Mani | 2006 | Low Risk | Medium Risk | Medium Risk | Low Risk |
| McCutchen | 1992 | Medium Risk | Medium Risk | Medium Risk | Medium Risk |
| MedProve | 2020 | Medium Risk | Medium Risk | Medium Risk | Medium Risk |
| Mossialos | 2016 | Medium Risk | Medium Risk | Medium Risk | Medium Risk |
| Nakamura | 2003 | High Risk | Medium Risk | High Risk | Medium Risk |
| Ruggieri | 2015 | Low Risk | Medium Risk | Medium Risk | Low Risk |
| Simpkin | 2017 | Low Risk | Medium Risk | Medium Risk | Low Risk |
| Srinivasan (a) | 2009 | Medium Risk | Medium Risk | Medium Risk | Low Risk |
| Srinivasan (b) | 2009 | Medium Risk | Medium Risk | Medium Risk | Medium Risk |
| Thompson | 2014 | Medium Risk | Low Risk | Medium Risk | Medium Risk |
| Treasury | 2004 | High Risk | Medium Risk | High Risk | High Risk |
| Tsui-Auch | 1998 | High Risk | Medium Risk | Medium Risk | High Risk |
| Wainberg | 1991 | Medium Risk | Medium Risk | Low Risk | Medium Risk |
| Young | 2017 | Medium Risk | Low Risk | Low Risk | Medium Risk |
| Yuan | 1990 | Low Risk | Low Risk | Low Risk | Medium Risk |

The 29 direct funding initiatives comprised 14 interventions to directly fund researchers, 5 interventions to upskill the workforce to enable more clinical trials, 4 initiatives to fund clinical trial infrastructure such as databases, 2 initiatives to fund patient costs, 2 initiatives to fund private companies' costs, as well as 2 advanced market commitments (Table 2).

**Table 2. Intervention types and forms of outcome assessment.**

| | Number of interventions of each type | Clinical trial activity outcome* | | | |
|---|---|---|---|---|---|
| | | Number of trials only | Expenditure on trials only | Expenditure & number of trials | Missing |
| Direct funding | 29 | 6 | 0 | 8 | 15 |
| Taxation policy | 14 | 1 | 5 | 5 | 3 |
| *Total* | *43* | *7* | *5* | *13* | *18* |

**Table 3. Characteristics of effects interventions in studies reporting outcome data.**

| | First author | Year of publication | Country/region | Taxation intervention | Further Information | Directly Funded Intervention | Further Information |
|---|---|---|---|---|---|---|---|
| | **Studies reporting outcomes for direct funding and taxation intervention** | | | | | | |
| 1, 2 | Austrade | 2018 | Australia | R&D Tax Credit / Offset | The R&D Tax Incentive gives companies with an annual turnover of <$20M: 43.5% tax credit, & >$20 million a 38.5% on eligible expenditure. One industry group asserts that the R&D tax credit is responsible for ~10% of Australian clinical trial activity [42] | Researcher Based Funding | Australian Government established a $20 billion Medical Research Future Fund for medical research two grants amounting to $614.2M [43] over 10 years aimed at specifically increasing clinical trial activity although to date only $212M has been distributed on 53 projects so far [43,44]. |
| 3, 4 | Chit | 2018 | USA | R&D Tax Credit / Offset | The Orphan Drug Act 1983 provided a 50% tax credit for expenditures incurred in the R&D of a rare-disease drug. Cumulative initiatives in the act led to a 69% increase of new clinical trials [22]. | Funding of Patient Participants | The Orphan Drug Act extended public health insurance (Medicare) to cover the costs of trials for patient participants. Yin [22] observed a marked increase in new clinical trials for drugs for rare disorders in the three years immediately after the ODA passed–however this reflected the cumulative impact of interventions in the ODA. |
| | **Studies reporting outcomes for direct funding alone** | | | | | | |
| 5 | Blume-Kohout | 2009 | USA | - | | Researcher Based Funding | The American Recovery and Reinvestment Act increased funding of $8.2 billion to fund extramural life sciences R&D resulting in a 3–6% increase in Phase I clinical trials |
| 6 | Blume-Kohout | 2012 | USA | - | | Researcher Based Funding | U.S. National Institutes of Health (NIH) grants researchers funding. A sustained 10% increase in targeted, disease-specific NIH funding yields approximately a 4.5% increase in the number of related drugs entering clinical testing (phase I trials) after a lag of up to 12 years |
| 7 | Chit | 2018 | USA | - | | Researcher Based Funding | Direct grants to researchers provided for in the Orphan Drug Act 1983. Cumulative initiatives in the act led to a 69% increase of new clinical trials [22]. |
| 8 | Chinnery | 2021 | UK | - | | Researcher Based Funding | Clinical Research Network which helped attract large private investments into COVID based clinical trials, the most successful of which was RECOVERY which enrolled up to 40% of all hospitalized COVID patients. |
| 9 | Davies | 2016 | UK | - | | Researchers Based Funding | Creation and expansion of National Institute for Health Research (NIHR) as a research funding body. This has led to rapid growth of clinical trials culminating in a 30% increase in trials over a 3-year period (between FY 2016/17–2018/19) [45]. |

(*Continued*)

**Table 3.** (Continued)

| | First author | Year of publication | Country/region | Taxation intervention | Further Information | Directly Funded Intervention | Further Information |
|---|---|---|---|---|---|---|---|
| 10 | De Padua Risolia | 2015 | Brazil | - | | Advanced Market Commitment | Brazilian Ministry of Health commits to fund vaccines and medicines through a foundation from phase 1 through to commercialization with clinical trials occurring within Brazil. The average cost of the clinical trials development in Brazil is now around 75%–80% of the related cost of US clinical trials and there has been a strong growth in the number of trials although this is difficult to directly link to this initiative [46]. |
| 11 | Iizuka | 2016 | Japan | - | | Funding of Patient Participants | Japanese Government offered financial support for the patient through patient cost sharing. Firm-sponsored new clinical trials increased by as much as 181% when covered by the policy [13]. |
| 12 | Kim | 2021 | USA | - | Researcher Based Funding | | Operation Warp Speed included $18B of US Govt funding of private companies for vaccine development and associated trials. This was a major contributing factor in the US' 1/3 of COVID-19 trials globally, (EU: 23%, China 5%) [47]. |
| 13 | MedProve | 2020 | South Korea | - | | Clinical Trial Infrastructure | Since its establishment the Korean Drug Development Fund (KDDF) has built 15 clinical trial centres. From 2007 to 2013, there was a 640% increase in the number of investigator-initiated clinical trials. |
| 14 | Ruggieri | 2015 | EU | | | Researcher Based Funding | €98.6M granted to researchers to investigate off-patent paediatric medicines. This represented 15% of all investments for research projects related to child health to conduct a total of 71 paediatric studies including 32 clinical trials, corresponding to an average of only 1.4 million euros for each study or trial. This investment increased the number of paediatric patients included in clinical trials in Europe by 23% over the same time period (2007 to 2011). |
| 15 | Srinivasan | 2009 | India | - | | Workforce | The government established training institutes to improve the clinical trial workforce. The paper points to cumulative efforts from several interventions that has led to "steady increases" in clinical trials primarily from the pharmaceutical industry. |
| 16 | Wainberg | 1991 | Canada | - | | Researcher Based Funding | $20.5 million was dedicated to a 5-year initiative to promote HIV and AIDS research; increased by $12 million in 1992–1993. A large amount of these funds were used to fund new clinical trials though no specifics were given. |
| | **Studies reporting outcomes for taxation alone** | | | | | | |

*(Continued)*

**Table 3.** (Continued)

| | First author | Year of publication | Country/ region | Taxation intervention | Further Information | Directly Funded Intervention | Further Information |
|---|---|---|---|---|---|---|---|
| 17 | ACoSaQiH | 2020 | South Korea | R&D Tax Credit / Offset | The government provides tax deductions for research and development costs and has established the Global Pharmaceutical Industry Development Fund through the Ministry of Health and Welfare for further incentives. From 2007 to 2013, there was a 50% increase in the number of sponsor-initiated commercial oncology trials, while investigator—initiated ones increased by 640% [48] although this is likely a cumulative result of several interventions. | - | |
| 18 | Choudhury | 2019 | India | R&D Tax Credit / Offset | Expenditure can be offset against other income for R&D on rare diseases by companies who have been pre-classified by the Department of Scientific and Industrial Research. Cumulative interventions have together been 'hugely successful' in encouraging additional clinical trials for orphan drugs. Three orphan medical products have been registered in India since the change in taxation treatments in 2012. | | |
| 19 | Choudhury | 2019 | India | Fees and Charges Exemptions | The Ministry of Health and Family Welfare has published the final version of New Drugs and Clinical Trials Rules, 2019 which waves all fees and charges for medicines affecting rare populations (1:500,000 people in India) [49] Three orphan medical products have been registered in India since the change in taxation treatments in 2012. | | |
| 20 | Hirst | 1991 | Australia | R&D Tax Credit / Offset | In 1985 Australia introduced one of the most generous R&D tax credits in the world at 150% for private sector expenditure. A subsequent review of this tax credit found it had inducement rates of 16.7% meaning that for every $10M of expenditure, it would induce a company to invest an additional $1.67M which is considered high [50]. The year following its introduction saw a sharp increase (close to double) in the number of PubMed clinical trial publications [51]. | | |
| 21 | Mani | 2006 | India | R&D Tax Credit / Offset | The explicit aim to grow clinical trials was given to support a 150% R&D Tax Credit that began in 2001/2. Over the next ten years clinical trials grew from 40–50 trials registered India to over 1850 trials registered in 2011 [52]. | | |

(Continued)

**Table 3.** (Continued)

| | First author | Year of publication | Country/region | Taxation intervention | Further Information | Directly Funded Intervention | Further Information |
|---|---|---|---|---|---|---|---|
| 22 | McCutchen | 1992 | USA | R&D Tax Credit / Offset | The Economic Recovery and Tax Act of 1981 provided for a 125% tax credit. The effectiveness of which was contested. At the high end, it was estimated that the credit stimulated between 15 to 36% while other studies have this as low as 0.6% | | |
| 23 | Mossialos | 2016 | China | R&D Tax Credit / Offset | Corporate Income Tax Law 2007/8 set R&D tax rates at 150% The paper points to massive growth of the CT industry 4 trials registered in 2001 to 497 in 2010 however much of this exponential growth took place before the tax law changes. | | |
| 24 | Mossialos | 2016 | China | Preferential Income Tax Rates | Reduced corporate tax rate by 15% for eligible companies. The effectiveness of which was the massive growth of the CT industry: 4 trials registered in 2001 to 497 in 2010, however much of this exponential growth took place before the tax law changes. | | |
| 25 | Yuan | 1990 | Japan | R&D Tax Credit / Offset | The Japanese government implemented a 120% tax credit on new R&D expenditure. Although clinical trials have grown significantly over time, it is difficult to attribute that growth to the tax credit. | | |

The 14 interventions targeting the taxation system comprised 11 interventions that made tax credits or offsets available for companies to stimulate research and development, 2 interventions where governments removed fees and charges usually applied to clinical trials and a single intervention that changed the tax rates applicable to firms that conducted clinical trials (Table 2). Characteristics of the interventions for which outcome data was available is outlined in Table 3, while the characteristics of the interventions for which no outcome data was available is outlined in Table 4.

## Effects of funding interventions on clinical trial activity

18 of the 41 interventions did not provide data on their impact but there were 6 interventions that reported on clinical trial numbers exclusively, 5 interventions that reported on clinical trial expenditure exclusively and 12 that reported on both. All 12 interventions that used direct funding reported positive outcomes and all 11 interventions that used a tax-based mechanism reported positive outcomes. The data qualifying the magnitude of impact were inconsistently reported and quantitative summary of effects was not possible.

**Direct funding interventions.** Targeted funding of researchers through special schemes administered by established funding bodies such as the United States' National Institutes of Health or the United Kingdom's National Institute for Health Research was one of the most commonly reported intervention types. A sustained 10% increase in targeted, disease-specific funding of researchers was estimated to yield a 3–6% increase in the number of drugs entering phase I clinical trials but only after a lag of up to 12 years and with no clear impact on later

**Table 4. Characteristics of interventions in studies that reported no outcome data.**

| | First author | Year of publication | Country/ region | Taxation intervention | | Directly Funded Intervention | |
|---|---|---|---|---|---|---|---|
| 26 | Cheng | 2007 | Australia | | | Researcher Based Funding | Cancer Australia committed $20 million between 2005–2009 to build the national capacity for clinical trials |
| 27 | Cheng | 2007 | China | | | Researcher Based Funding | An increased allocation of funding (typically HK$1M over three years) to university researchers aimed explicitly at cover the costs of clinical trials. |
| 28 | Cheng | 2007 | Japan | | | Researcher Based Funding | Japanese Governments fund oncology research, researchers and specifically clinical trials through a variety of mechanisms and bodies. |
| 29 | Cheng | 2007 | Singapore | | | Clinical Trial Infrastructure | Agency for Science and Technology Research is a funding agency tasked with coordinating Singapore's cancer research. They have a 5-year budget of S$75 million that is to be used to fund the development of clinical trial research infrastructure focused on cancer. |
| 30 | Christakis | 1989 | USA | | | Advanced Market Commitment | If developed, it guaranteed purchase of 500,000 units of an AIDS/HIV vaccine at $20 per unit. |
| 31 | Christakis | 1989 | USA | | | Funding for Private Company | A direct grant of US$6M to private companies for funding of clinical trials. |
| 32 | Hudson | 2016 | USA | | | Workforce | Good Clinical Practice training for investigators and NIH staff responsible for conducting or overseeing clinical trials. |
| 33 | Nakamura | 2003 | Japan | | | Workforce | A 1998 Government policy established a support system for clinical trials including the contribution of clinical research coordinators to provide consistent and flexible management across a network of trial sites. |
| 34 | Simpkin | 2017 | EU | | | Funding for a Private Company | InnovFin Infectious Diseases is a financial instrument developed by the European Commission & European Investment Bank. It offers loans between €7.5-€75M to develop innovative vaccines, drugs, medical & diagnostic devices, for combatting infectious diseases. It is a risk-sharing initiative, as the loan is only paid back if the project is successful |
| 35 | Simpkin | 2017 | EU | | | Researcher Based Funding | Funding from programs known as Horizon, The Directorate-General for Research and Innovation, European Commission is one of the largest funding bodies supporting the R&D of antibiotics, alternative medicines, and diagnostic tools at around €1B and almost 6B for Oncology research |

*(Continued)*

**Table 4.** (Continued)

| | First author | Year of publication | Country/ region | Taxation intervention | | Directly Funded Intervention | |
|---|---|---|---|---|---|---|---|
| 36, 37 | Srinivasan | 2009 | India | Fees and Charges Exemptions | Clinical trials have been exempted from sales tax. Import duty has been lifted on clinical trial supplies and permission for export of clinical trial specimens will be granted at the same time as the protocol is approved. | Clinical Trial Infrastructure | Hospital departments running trial sites gain additional equipment and the salaries of junior/additional investigators (paid for by the trial sponsor for the duration of the trial). These additional resources are often deployed to other trials. |
| 38, 39 | Thompson | 2014 | Canada | R&D Tax Credit / Offset | The province also seeks to lower costs for trialists through its R&D tax incentive program, available to qualified businesses of any size, and applies to a range of eligible costs that is broader than that available in the United States. | Researcher Based Funding | Clinical Trials Ontario has set up an investment fund that helps cover the costs of researchers' clinical trials in the province. |
| 40 | Treasury | 2004 | Africa | | | Clinical Trial Infrastructure | 400 million provided for clinical trial capacity building through trusts and research centres |
| 41, 42 | Tsui-Auch | 1998 | Singapore | R&D Tax Credit / Offset | The Economic Development Board has provided R&D tax credits as subsidies and incentives to both local and foreign firms. | Workforce | Quintiles East Asia Pty Ltd. collaboration with the National University of Singapore to establish a regional clinical research program that aimed to train ~300 professionals in procedures that meet up to Good Clinical Practice standards |
| 43 | Young | 2017 | Africa | | | Workforce | Funding has been allocated from a variety of trusts and funding programs to boost the capacity of the African researcher through Masters and PhD programs. |

stage phase II or III studies [15]. Greater impacts may be achieved by funding targeted at specific under-researched areas such as rare diseases [16,21,22] where increases in new clinical trials of up to 69% have been reported. The largest effects were observed from multi-modal interventions targeting the same diseases areas under the United States Orphan Drug Act, which included direct funding of researchers and trial participant costs as well as an R&D tax credit for participating corporations. Similarly, Canada achieved a large increase in trials addressing human immune deficiency virus in the early 1990s following a dedicated $20.5M funding tranche provided direct to researchers [24]. Less focused initiatives, such as the establishment of a new funding route through the United Kingdom National Institute for Health Research (NIHR), have also driven substantial growth. As well as providing significant additional funding to researchers, the establishment of the NIHR funding mechanisms is credited with attracting new funding from commercial companies (£4.4B between FY 2016–19) leading to a 30% increase in the number of clinical trials done over that period [45].

Governments have also direct-funded a wide range of initiatives to enable the better conduct of clinical trials. The Korean Drug Development Fund established 15 purpose-built clinical trial centres attached to major teaching hospitals operating under the supervision of the Korea National Enterprises for Clinical Trials that also runs a training academy for professionals

involved in clinical trials. From 2007 to 2013, there was a 50% increase in the number of sponsor-initiated commercial oncology trials, while investigator-initiated trials increased by 640%.

The COVID_19 pandemic led to an unparalleled increase in investment in research and development with a corresponding increase in clinical trials at each phase. The United States' response to the pandemic named 'Operation Warp Speed' included $18 billion (USD) of government funding of private companies for vaccine development and associated trials. This was a major contributing factor in the that country attracting 33% of all COVID_19 trials globally compared to the EU at 23%, and China at 5% [47]. Agarwal and Gaule found that the U.S. and Chinese vaccine candidates were on average 2 months faster to move to a pre-clinical phase than vaccines from other countries. This crucial boost in speed was 'possibly due to greater provision of early-stage incentives by the policy response in these countries, including through programs such as Operation Warp Speed' [47]. In the United Kingdom the Clinical Research Network helped attract large private investments into COVID based clinical trials, the most successful of which was RECOVERY which enrolled up to 40% of all hospitalized COVID patients [39].

**Taxation policy interventions.** R&D tax credits are one of the most widely deployed interventions used by governments to attract foreign investment in clinical trials, comprising 9/10 of the taxation policy interventions with outcome data that were identified.

Australia was one of the first jurisdictions to introduce a tax credit for clinical trials in 1985 rebating 150% for private sector expenditure. An inducement rate of 16.7% was reported meaning that for every $10M of scheduled corporate expenditure, a participating company invested an additional $1.67M. Both the 150% rebate and the ensuing 16.7% inducement rate are considered high [50]. The rebate was subsequently lowered, removed altogether and recently reinstated such that small-to-medium enterprises with an annual turnover of <$20M receive a 43.5% tax credit on eligible expenditure versus a 38.5% tax credit for larger companies. An industry group report asserts that the R&D tax credit is responsible for about 10% of Australian clinical trial activity [42].

India has also deployed an R&D tax credit to boost clinical trials activity commencing in 2001 at 150%. Due to its success, the rate was increased to 200% between 2010–2017 [53] but was later reduced due to concerns about effectiveness when national research spending shrank from 0.83% to 0.63% of Indian gross domestic product between 2011 and 2015 [54].

The United States' R&D tax credit was first implemented in 1982 as part of The Economic Recovery and Tax Act passed the year prior and provided a credit of 125% for eligible items of expenditure. The effectiveness of the measure has been contested with proponents arguing it contributed between 15 and 36% of clinical trial activity, while others have estimated this metric to be as low as 0.6% [32].

## Discussion

The literature indicates that governments have a clear opportunity to stimulate clinical trial activity through fiscal interventions, though evidence about the magnitude of the effect and the comparative effectiveness of different strategies is limited. The most widely reported method was direct support of clinical trial components through funding of researchers, companies, infrastructure, workforce or the trial participants themselves. The other was through interventions that targeted the taxation regimes applicable to companies performing clinical trials.

Interventions that directly funded specific trials or a particular piece of required infrastructure were able to be delivered precisely to address specific deficiencies with fairly easily measured and reported outcomes. As a consequence, and perhaps unsurprisingly, there is little doubt that targeting funding to clinical trials in an under-researched disease area, or

supporting the growth of a deficient workforce sector, can deliver a rapid and direct return on investment. Less certain are the long-term effects of direct funding mechanisms, since while workforce development programs might be anticipated to provide a sustained impact beyond the funding period, it is easy to imagine that the research workforce might rapidly switch their efforts towards other non-trial opportunities once a specific granting scheme discontinued.

The interventions targeting taxation policies were focused on providing research and development credits or offsets, whereby potential government tax revenue was forgone if a company invested in relevant areas. They are designed to facilitate a private sector response (contrasted with directly funded initiatives that were mostly targeted towards public entities). Typically, clinical trials were only one part of eligible corporate expenditure, and while the literature found these interventions positive, they were not a precise a tool by which to target clinical trial activity. Rather, they boosted multiple aspects of research and development, many of which were only peripherally associated with clinical trials. As such, taxation policies may be a more expensive intervention for governments seeking to increase clinical trial numbers or lower the costs associated with doing clinical trials, though broader benefits across the research sector may accrue. In one intervention in India, the government removed import duties only on items required for use in clinical trials, [5,37] which enabled more specific targeting of the intervention. Likewise, in another Indian government intervention, more generous tax exemptions were provided specifically for trials relating to orphan medicines or rare diseases where the nature of the patient population makes economies of scale impossible to achieve [29]. This strategy provided for a more targeted application of the taxation benefits. A final approach to targeting of preferential income tax rates was to make only a highly selected subset of companies eligible [36].

Developing country nations have achieved a major increase in their share of the clinical trial market in recent times. India, for example, grew its portion of global clinical trials from 0.9 per cent in 2008 to 5 per cent in 2013 and China has experienced similar expansion. At the same time, the share of clinical trial activities done in the United States and other developed countries has been declining as research organisations seek to take advantage of lower costs [55]. Likely as a reflection of this trend, 29 of the 41 fiscal interventions identified by this systematic review were done by developed countries seeking to lower their costs and boost their competitiveness.

## Strengths and limitations

This review benefitted from our systematic search of the literature done to try and capture all relevant information. The algorithms used by search engines can weight results towards user characteristics such as geography and language and this may have mitigated against the detection of reports from countries such as China and Korea—two markets that have significant clinical trial activity but for which relatively few search results were returned. Additionally, most of the included studies were set in English-speaking jurisdictions and this may have been due to the exclusive use of English search terms and the algorithms.

It is also possible that the search results were influenced by publication bias, which it was not possible to formally test for, given the limited quantitative data available. The literature did not identify any adverse or null outcomes but there were 15 studies with "missing" results and interventions that did not achieve stated objectives or had adverse effects on clinical trial activity may be over-represented in this subset. The inclusion of grey literature ensured that more relevant data were included but the quality of reporting was more varied, and this presented analytic challenges [20,56]. It was not possible to search every possibly relevant result returned from the grey literature searches because of the very large numbers obtained and we took a

pragmatic approach to terminating review once the proportion of hits fell substantially. The standardised and duplicated extraction of information from the identified reports served to maximize the quality of the data that was available. The outcome data about effects on clinical trial activity outcomes were described inconsistently using different metrics and a quantitative summary was not possible as a consequence, though the systematic tabulation provided for a high-quality narrative review [12].

The studies came from only a relatively small number of jurisdictions that are not representative of the globe though there was a mix of higher and lower-income countries included. As such there is some uncertainty about the extent to which the main conclusions are generalisable though it is likely that key themes, such as the specificity of effects of direct funding versus tax breaks, will be common across multiple settings.

Although the results of the interventions are universally positive where reported, these studies do not assess the opportunity costs. It is possible, for example, that the relatively high cost of the research and development tax credits could have delivered better value for money if expended on direct grants to researchers, or specific infrastructure to enable more trials to be performed. Similarly, the review was unable to measure and compare the quantitative effects of the different fiscal intervention types because the available data were too few and too diverse. It is also possible that the level of background research activity in a jurisdiction might modify the impact of an intervention–for example, interventions may be more effective if done in a setting with extensive infrastructure or may be viewed as delivering only marginal incremental change if done in a jurisdiction where large existing direct or indirect funding schemes are already operating.

## Conclusion

Our data show that governments can achieve enhanced clinical trial activity by direct funding initiatives to boost clinical trials or by targeting the tax treatments of the companies conducting the research. Where governments achieve greater clinical trial activity there is also a reasonable expectation that the research sector, the health system, the community, and the economy will benefit and there is a good likelihood that the costs will be offset.

## Supporting information

**S1 Checklist. Prisma checklist.**
(DOCX)

**S1 Fig. Search strategy flowchart.** S1 Fig shows the search process that researchers SC and ER undertook.
(TIF)

**S2 Fig. Direct funding and taxation policy interventions by government.** S2 Fig demonstrates the geographical locations that each intervention took place in.
(TIF)

**S1 Appendix. Studies reporting direct funding initiatives or taxation policy interventions.** Graph shows the year that each identified paper was published.
(DOCX)

## Author Contributions

**Conceptualization:** Sam Crosby, Stephen Jan, Bruce Neal, Richard Holden.

**Data curation:** Sam Crosby, Esther Rajadurai.

**Formal analysis:** Sam Crosby, Bruce Neal.

**Investigation:** Sam Crosby, Esther Rajadurai.

**Methodology:** Sam Crosby, Stephen Jan, Bruce Neal.

**Project administration:** Sam Crosby, Esther Rajadurai.

**Supervision:** Stephen Jan, Bruce Neal, Richard Holden.

**Writing – original draft:** Sam Crosby.

**Writing – review & editing:** Sam Crosby, Stephen Jan, Bruce Neal, Richard Holden.

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
