## [Decision Letter · Decision Letter 0]

19 Aug 2022

PONE-D-22-13387The effects on clinical trial activity of direct funding and taxation policy interventions made by government: a systematic reviewPLOS ONE

Dear Dr. Samuel Patrick Crosby,

Thank you for submitting your manuscript to PLOS ONE. After careful consideration, we feel that it has merit but does not fully meet PLOS ONE’s publication criteria as it currently stands. Therefore, we invite you to submit a revised version of the manuscript that addresses the points raised during the review process.

We look forward to receiving your revised manuscript.

Kind regards,

María del Carmen Valls Martínez, Ph.D.

Academic Editor

PLOS ONE

Journal Requirements:

2. We note that Figure 2 in your submission contain map/satellite images which may be copyrighted. All PLOS content is published under the Creative Commons Attribution License (CC BY 4.0), which means that the manuscript, images, and Supporting Information files will be freely available online, and any third party is permitted to access, download, copy, distribute, and use these materials in any way, even commercially, with proper attribution. For these reasons, we cannot publish previously copyrighted maps or satellite images created using proprietary data, such as Google software (Google Maps, Street View, and Earth). For more information, see our copyright guidelines: http://journals.plos.org/plosone/s/licenses-and-copyright.

a) You may seek permission from the original copyright holder of Figure 2 to publish the content specifically under the CC BY 4.0 license.  

3. We note you have included tables to which you do not refer in the text of your manuscript. Please ensure that you refer to Tables 2 and 3 in your text; if accepted, production will need this reference to link the reader to the Table.

4. Please upload a copy of Figure 3, to which you refer in your text on page xx. If the figure is no longer to be included as part of the submission please remove all reference to it within the text.

Reviewers' comments:

Reviewer's Responses to Questions

**Comments to the Author**

1. Is the manuscript technically sound, and do the data support the conclusions?

Reviewer #1: Yes

Reviewer #2: Yes

2. Has the statistical analysis been performed appropriately and rigorously? 

Reviewer #1: Yes

Reviewer #2: Yes

3. Have the authors made all data underlying the findings in their manuscript fully available?

Reviewer #1: Yes

Reviewer #2: Yes

4. Is the manuscript presented in an intelligible fashion and written in standard English?

Reviewer #1: Yes

Reviewer #2: Yes

5. Review Comments to the Author

Reviewer #1: -The primary outcomes of interest were the number of clinical trials and the expenditure on clinical trials but data about other evaluations were also collected;

- All 25 of the studies for which data were available reported a positive impact on numbers and/or expenditure on clinical ;

- Direct funding may provide a more immediate and tangible return on investment than tax rebates;

- Estimates of the magnitude of effects of interventions were reported inconsistently, varied substantially, and could not be synthesised quantitatively, though targeted direct funding interventions appeared to be associated with more immediate impact on clinical trial activity.

Reviewer #2: According to my opinion article is focused on very interesting and actual topics. I really appreciate the quality of the research and also very interesting conclusions. In the text, I found a few grammatical mistakes. The authors used a lot of quality sources. Some of them are older. I would suggest that authors could try to find new sources.

In general, I recommend publishing a paper.

6. PLOS authors have the option to publish the peer review history of their article (what does this mean?). If published, this will include your full peer review and any attached files.

Reviewer #1: No

Reviewer #2: No

---

## [Author Response · Author response to Decision Letter 0]

21 Aug 2022

Dear Editors,

Thank you so much for your comments and suggestions, as well as those of the reviewers. I have included a more comprehensive response attached to this submission but below is a summary of the changes made. 

1. Completed

2. Figure 2 was created using the map tool on a spreadsheet on a fully licenced version of Microsoft excel, (as was appendix 1). As such it complies with Microsoft’s copyright permissions: “You may sell a spreadsheet, database, or PowerPoint deck you made using Microsoft software. The spreadsheet, database, or PowerPoint deck must be created using legitimate, licensed Microsoft software.” Please see: https://www.microsoft.com/en-us/legal/intellectualproperty/copyright/permissions

3. Amended. Please see the bottom of page 6. The heading has been amended as follows: “Characteristics of the interventions and the available outcome data (Table 2 & 3)” and at the bottom of that section, the following sentence has now been inserted: “Characteristics of the interventions for which outcome data was available is outlined in Table 2, while the characteristics of the interventions for which no outcome data was available is outlined in Table 3.”

4. All references to Figure 3 have now been deleted. 

5. The reference list has been reviewed and substantial changes made to the formatting throughout to keep in line with PLOS ONE’s publishing style. 

6. Responding to reviewer two's comments: The paper has been reviewed for grammatical mistakes and minor revisions made. All references have been reviewed. The older references that are referred to were captured in the search processes documented in the paper. Removing these references would require structural changes to the search and would alter the paper’s findings. All other references – not found in the systematic review – are current. 

Thank you

---

## [Decision Letter · Decision Letter 1]

23 Aug 2022

The effects on clinical trial activity of direct funding and taxation policy interventions made by government: a systematic review

PONE-D-22-13387R1

Dear Dr. Samuel Patrick Crosby,

We’re pleased to inform you that your manuscript has been judged scientifically suitable for publication and will be formally accepted for publication once it meets all outstanding technical requirements.

Kind regards,

María del Carmen Valls Martínez, Ph.D.

Academic Editor

PLOS ONE

Additional Editor Comments (optional):

Reviewers' comments:

Reviewer's Responses to Questions

**Comments to the Author**

1. If the authors have adequately addressed your comments raised in a previous round of review and you feel that this manuscript is now acceptable for publication, you may indicate that here to bypass the “Comments to the Author” section, enter your conflict of interest statement in the “Confidential to Editor” section, and submit your "Accept" recommendation.

Reviewer #2: All comments have been addressed

2. Is the manuscript technically sound, and do the data support the conclusions?

Reviewer #2: Yes

3. Has the statistical analysis been performed appropriately and rigorously? 

Reviewer #2: Yes

4. Have the authors made all data underlying the findings in their manuscript fully available?

Reviewer #2: Yes

5. Is the manuscript presented in an intelligible fashion and written in standard English?

Reviewer #2: Yes

6. Review Comments to the Author

Reviewer #2: I recommend acceptance of the paper. All suggestions were implemented in the text. Paper is in good quality.

7. PLOS authors have the option to publish the peer review history of their article (what does this mean?). If published, this will include your full peer review and any attached files.

Reviewer #2: No

---

## [Editor Report · Acceptance letter]

30 Aug 2022

PONE-D-22-13387R1 

The effects on clinical trial activity of direct funding and taxation policy interventions made by government: a systematic review 

Dear Dr. Crosby:

I'm pleased to inform you that your manuscript has been deemed suitable for publication in PLOS ONE. Congratulations! Your manuscript is now with our production department. 

Kind regards, 

on behalf of

Dr. María del Carmen Valls Martínez 

Academic Editor

PLOS ONE